# Radiologic Evaluation of Uterine Vasculature of Uterus Transplant Living Donor Candidates: DUETS Classification

**DOI:** 10.3390/jcm11154626

**Published:** 2022-08-08

**Authors:** Jakub Kristek, Liza Johannesson, Matthew Paul Clemons, Dana Kautznerova, Jaroslav Chlupac, Jiri Fronek, Giuliano Testa, Gregory dePrisco

**Affiliations:** 1Department of Transplantation Surgery, Institute for Clinical and Experimental Medicine, Videnska 1958/9, 140 21 Prague, Czech Republic; 2Department of Anatomy, Second Faculty of Medicine, Charles University, V Uvalu 84, 150 06 Prague, Czech Republic; 3Annette C. and Harold C. Simmons Transplant Institute, Baylor University Medical Center, 3410 Worth St Ste 950, Dallas, TX 75246, USA; 4Department of Obstetrics and Gynecology, Baylor University Medical Center, 3500 Gaston Ave, Dallas, TX 75246, USA; 5Department of Radiology, Baylor University Medical Center, 3500 Gaston Ave, Dallas, TX 75246, USA; 6Department of Diagnostic and Interventional Radiology, Institute for Clinical and Experimental Medicine, Videnska 1958/9, 140 21 Prague, Czech Republic; 7First Faculty of Medicine, Charles University, Katerinska 1660/32, 121 08 Prague, Czech Republic

**Keywords:** blood supply, computed tomography angiography, diagnostic imaging, donor selection, magnetic resonance angiography, transplantation, uterus

## Abstract

Uterus transplantation is a treatment solution for women suffering from absolute uterine factor infertility. As much as 19.5% of uterus-transplanted patients underwent urgent graft hysterectomy due to thrombosis/hypoperfusion. The necessity to identify candidates with high-quality uterine vasculature is paramount. We retrospectively evaluated and compared the imaging results with actual vascular findings from the back table. In this article, we present a novel radiologic grading scale (DUETS classification) for evaluating both uterine arteries and veins concerning their suitability for uterus procurement and transplantation. This classification defines several criteria for arteries (caliber, tapering, atherosclerosis, tortuosity, segment, take-off, and course) and veins (caliber, tapering, plethora, fenestrations, duplication/multiplicity, dominant route of venous drainage, radiologist’s confidence with imaging and assessment). In conclusion, magnetic resonance angiography can provide reliable information on uterine venous characteristics if performed consistently according to a well-established protocol and assessed by a dedicated radiologist. The caliber of uterine arteries seems to be inversely related to the time passed since the last delivery. We recommend that the radiologist comments on the reliability and confidence of the imaging study. It cannot be over-emphasized that the most crucial aspect of surgical imaging is the necessity of high-quality communication between a surgeon and a radiologist.

## 1. Introduction

Uterus transplantation (UTx) is the only treatment for women affected by absolute uterine-factor infertility (AUFI) who wish for gestational parenthood [1,2,3,4,5]. While inflow to the human native uterus is anatomically provided by three pairs of arteries (ovarian, uterine, vaginal), the inflow to a uterine graft is delivered by two uterine arteries (UA) only. The outflow of the uterine graft is provided by two to four venous anastomoses using either superior uterine veins (SUV) and/or inferior uterine veins (IUV) [4]. The quality and caliber of uterine graft vessels are critically important because thrombosis of any of the graft’s supplying vessels may lead to graft loss, which is apparent in the analysis of a cohort of transplanted patients. Urgent graft hysterectomy was performed in 12 of 51 transplanted cases (23.5%) [6]. As much as 83% (10/12) of the graft failures were due to thrombosis or hypoperfusion, i.e., as much as 19.5% (10/51) of uterus-transplanted patients underwent urgent graft hysterectomy due to thrombosis or hypoperfusion [6]. Similarly, poor quality of UA and consequential hypo-perfusion were suggested as potential causal factors for early graft failure [7,8,9].

One of the biggest challenges in selecting potential uterus donors is to identify and exclude candidates with either structural abnormalities or with low-quality uterine vasculature. UTx is a non-life-saving procedure. Hence, it is paramount to keep the morbidity of the donor and recipient as low as possible. Currently, only two studies focus on the radiologic evaluation of uterine vasculature [7,10].

The primary objective of this paper is to propose radiologic criteria for the evaluation of uterine arteries and veins and to analyze our cohort using this classification.

## 2. Materials and Methods

### 2.1. Description of the Studies

This article is a retrospective study of radiologic evaluation of uterine arteries and veins of uterine living donor (LD) candidates. This study compares the imaging results with actual vascular findings from the back table. It was conducted as a part of two clinical trials: (i) the Dallas Uterus Transplant Study (DUETS) Clinical Trial; and (ii) the Czech UTx trial. DUETS was conducted at Annette C. and Harold C. Simmons Transplant Institute, Baylor University Medical Center, Dallas, TX, USA. The Czech UTx trial was conducted at the Institute for Clinical and Experimental Medicine, Prague, Czech Republic. DUETS is registered at ClinicalTrials.gov (NCT02656550). The Czech UTx trial is registered at ClinicalTrials.gov (NCT03277430). Both centers’ Institutional Review Board approved these clinical trials. Informed consent was obtained from all study participants. Both trials are compliant with the guidelines of the Declaration of Helsinki.

### 2.2. Patients Selection

The baseline evaluation of all LD 35 candidates was performed by abdominal ultrasound to exclude contraindication such as intraabdominal masses, polyps, adhesions, adenomyosis, etc. Four candidates were excluded due to findings on ultrasound (fibroids, adenomyosis), and one was excluded based on psychiatric conditions. Thirty included candidates were further evaluated with computed tomography angiography (CTA) with/without magnetic resonance angiography (MRA). Six out of 30 (20%) LD candidates were excluded for various reasons (diminutive caliber of veins, severe atherosclerosis, abnormal course, or take-off of UA). Due to the protocol change in the course of the DUETS, 14 of 24 included (58.3%) LD were also evaluated with MRA. All six LD candidates in the Czech trial were assessed with CTA only. None of the Czech candidates was excluded based on the results of the preoperative CTA evaluation. However, in one case of the Czech trial, a procured graft was contraindicated because of poor venous outflow options diagnosed first on the back table (not identified by preoperative CTA). None of the included LDs suffered from systemic lupus erythematosus or any other systemic disease. Twenty-three of 24 (95.8%) LDs were non-smokers. One LD had a history of smoking (35 pack years). None of the recipients had any prothrombotic mutation or any medical history that affects thrombosis.

### 2.3. Data Collection and Imaging Protocol

We retrospectively evaluated radiologic studies of 24 LD candidates: 10 LD candidates underwent CTA only, and 14 LD candidates underwent both CTA and MRA. All studies were retrospectively evaluated by a single dedicated radiologist (Gregory dePrisco). The retrospective evaluation of reports of uterine procurement, back table, and transplant procedure was performed by Jakub Kristek. All discrepancies were discussed with co-authors (Gregory dePrisco, Liza Johannesson, Jiri Fronek) until a unanimous agreement was reached. The imaging protocol for CTA and MRA evaluation of uterine LD has been published previously [10] (Appendix A).

## 3. Results

### 3.1. DUETS Radiologic Classification of Uterine Vasculature

Dr. Gregory dePrisco has introduced a new classification (DUETS classification) to produce a scale for evaluating the quality of both uterine arteries and veins. This classification focuses on the radiologic grading of uterine vasculature concerning its suitability for both uterine procurement and transplantation. This classification defines several criteria for arteries (caliber, tapering, atherosclerosis, tortuosity, segment, take-off, and course) and veins (caliber, tapering, plethora, fenestrations, duplication/multiplicity, dominant route of venous drainage, radiologist’s confidence with imaging and assessment). The details of the classification can be found in Table 1. Many examples of pictures of arteries and veins graded according to the DUETS classification can be found in Supplementary Appendix A.

### 3.2. Characteristics of Living Donor Candidates

The study cohort comprised 34 uterine LD candidates aged 30–58 years (median 38.5, 25th percentile 33.25, 75th percentile 46.5). Ten candidates were excluded due to contraindication (Figure 1). The body mass index ranged between 19 and 36 kg/m^2^ (mean 24.5). The mean number of deliveries of LD candidates was 2.4 (range 0–7). The mean time between the donor’s last delivery and radiologic evaluation was 7.9 years (from 8 months to 35 years). The enrollment and evaluation of LD candidates took 49 months, i.e., from 1 January 2016 to 31 January 2020.

### 3.3. Evaluation of Vessels of the Cohort of LDs

#### 3.3.1. Arteries

A retrospective evaluation of arterial diameters was performed. As reported by Leonhardt in 2021 [7], arterial diameters of 1 mm or greater seem to be adequate for transplant. We correlated the uterine artery’s luminal diameter with the graft thrombosis frequency. Luminal diameters were evaluated at the vessel origin and midsegment and classified as grade A if the diameter was greater than or equal to 2 mm, grade B if the diameter was greater than 1 but less than 2 mm, and grade C if 1 mm or less. Based on CTA findings in our cohort, the frequency of A, B, and C-graded arteries were 22.9%, 52.1%, and 25%, respectively. The tapering of the artery near the vessel’s take-off, thought to be a consequence of the obliterated umbilical artery, was described in 33 of 48 arteries (68.7%), with the minus designation applied after the vessel grading when tapering was observed. No atherosclerosis was described in 31 of 48 arteries (64.6%). Mild, moderate, and severe atherosclerosis was described in 11 (22.9%), 6 (12.5%), and none (0%) of 48 arteries, respectively. Mild, moderate, and severe tortuosity of UA was observed in 38 (79.2%), 9 (18,8%), and 1 (2%) of 48 arteries, respectively. The average length of the segment between the posterior division of the internal iliac artery (IIA) and the UA’s take-off was 6.3 mm on the right (SD ± 3.1, range 3–13) and 6.9 mm on the left (SD ± 4.1, range 0–13). The period from the last delivery to donation was 3.1 (*n* = 10, SD ± 2.1, range 1.2–7) in A-graded, 14.9 (*n* = 26, SD ± 8.6, range 3–35) in B-graded, and 13.4 years (*n* = 12, SD ± 8.6, range 0.7–35) in C-graded donors (Table 2). Similarly, the average number of deliveries was 3.7 (*n* = 10, SD ± 1.9, range 2–7) in A-graded, 2.1 (*n* = 26, SD ± 0.8, range 0–3) in B-graded, and 2.4 (*n* = 12, SD ± 1.1, range 1–4) in C-graded donors (Table 2). Two cases demonstrated a variant uterine arterial supply, one with bilateral ovarian arterial as the dominant supply and the other duplicated left UAs.

#### 3.3.2. Veins

A retrospective evaluation of venous diameters was performed. In this paper, we use the terminology defined by the guidelines for standardized nomenclature and reporting in uterus transplantation [4]. According to these guidelines, inferior uterine vein (IUV) is a term for what used to be called a uterine vein; superior uterine vein (SUV) is a term for what used to be called a utero-ovarian vein. Luminal diameters were evaluated at the vessel origin and midsegment. It was classified as grade A if the diameter was greater than or equal to 6 mm, grade B if the diameter was between 4 and 6 mm, and grade C if it was less than 4 mm. The number of veins described by CTA correlated with back-table findings in 48.6% of vessels, while MRA evaluation correlated in 72.7% of veins. The caliber of veins described by CTA correlated with back-table findings in 34.3% of vessels, while MRA evaluation correlated in 60.4% of veins (Table 3). The dominant route of venous drainage via left-sided veins (left IUV, left SUV) was present in 29%, drainage via both SUVs (right SUV, left SUV) in 17% of cases, and no apparent dominance was found in 25% of cases. MRA revealed a higher rate of correct venous criteria identification than CTA. In both cases lost due to venous thrombosis (5 venous anastomoses), there were A/A minus-graded veins in 20% (*n* = 1), B/B minus-graded veins in 0% (*n* = 0), and C/C minus graded veins in 80% of cases (*n* = 4) (Table 4). The rate of the radiologist’s confidence with the assessment of veins was 93% using MRA vs. 21% using CTA.

#### 3.3.3. Correlation of Imaging and Back Table Findings

The absence of a back-table report (either lost or not written at all) occurred in 8/24 cases (33.3%), including five cases in the Czech cohort (5/6) and three cases in the Dallas cohort (3/18). Incompleteness or ambiguity of description of the diameter of vessels in a back-table report occurred in 43.8% of existing back-table reports.

#### 3.3.4. Correlation of Vasculature and Recipient Outcome

The total graft survival at 1 year was shown in 17 of 23 (73.9%) transplanted uteri. Six grafts of 23 (26.1%) were lost due to thrombosis. In four cases (eight arteries) that developed arterial thrombosis/ischemia, the number of A/A-minus, B/B-minus, and C/C-minus arteries were 0 (*n* = 0), 62.5% (*n* = 5), and 37.5% (*n* = 3), respectively (Table 4). In other words, no grafts lost due to arterial thrombosis were supplied by an A-minus-graded artery. Contrarily, recipients with viable grafts without/with live births had C/C-minus arteries in 16.7% (*n* = 2) and 22.7% (*n* = 5), respectively.

#### 3.3.5. Live Birth

At the time of this report, 12 recipients have delivered at least one child. The frequency of A/A-minus, B/B-minus, and C/C-minus-graded arteries in this cohort is 6/24 (25%), 12/24 (50%), and 6/24 (25%), respectively. The frequency of A/A-minus, B/B-minus, and C/C-minus-graded veins in this cohort is 3/34 (8.8%), 19/34 (55.9%), and 12/34 (35.3%), respectively.

## 4. Discussion

There is no existent grading system for radiologic assessment of uterine arterial and venous anatomy concerning its suitability for potential UTx. Although UTx is a life-giving and life-promoting procedure, it is not life-saving. It is crucial to do our best to exclude marginal grafts and eliminate the unnecessary risk of morbidity, including the risk of thrombosis of the graft. We believe that precise preoperative assessment of uterine vessels might play a significant role in decision-making. The parameters of this classification are arbitrarily chosen, yet they might evolve in the future if some of them become redundant. We believe that stratification of arteries and veins used in a particular procedure (e.g., A-graded artery) could also help in a retrospective evaluation of the outcome of a cohort.

Of note, the caliber of UA seems to be inversely related to the time passed since the last delivery. In the study cohort, many donors with A-graded arteries donated their uterus after a shorter period than donors with B and C-graded arteries. This observation could be explained by partly residual uterine arterial and venous vasculature hypertrophy, which occurs throughout pregnancy. The question is whether this could be of any use in the planning of organ recovery. Six cases of graft thrombosis occurred in our cohort of 23 LD UTx recipients. None of the thrombosed grafts had A-graded arteries (Table 4). However, we believe that the data set is too small to draw any conclusions on how significant the time factor is compared to other factors for the quality of uterine blood vessels.

Uterine retrieval can be performed either open, laparoscopic, or robotic. All Czech LD uterine retrievals (six cases) were performed open (median 5 h 51 min). In DUETS, open LD uterine retrievals were performed in 13 cases (median 6 h and 27 min), and the robotic uterine retrievals were performed in eight cases (median 10 h and 46 min). Although robotic retrievals lasted longer than open procedures, no differences were observed in recipient outcomes.

So far, only two studies have focused on the preoperative radiologic evaluation of uterine vasculature [7,10]. The study of Leonhardt et al. compares CTA vs. digital subtraction angiography vs. MRA for the visualization of UAs of LD UTx candidates. They concluded that MRA should be the initial and the only modality in most cases. In a minority of cases when MRA fails to visualize the UAs satisfactorily, CTA would be performed [7]. Their study does not address the visualization of uterine venous drainage. We believe, as has already been published in the article of Mahmood et al. [10], that CTA and MRA play rather complementary roles in the visualization of uterine arteries and veins. CTA provides a higher spatial resolution of UAs and can detect calcified plaques more reliably. MRA, as we have shown, can yield more accurate results in the evaluation of uterine veins.

### 4.1. Technical Aspects

For nearly all arteries evaluated, CTA accurately depicted the arterial anatomy. However, in exceptional cases, CTA may yield poor images of UA, possibly due to spasm or arterial pulsation. For instance, the left UA in DUETS LD number 14 was graded C minus with CTA, i.e., overall diminutive and gaining in diameter over its course towards the uterus. However, there was a discrepant finding on MRA which revealed a perfectly normal appearance of the vessel (including its caliber), which was also confirmed intraoperatively. We assume the deceptive appearance on CTA may be attributed to the hyperdynamic pulsation of the artery. On the other hand, MRA with the venous phase highly facilitates the evaluation of IUV and SUV because of its superior contrast resolution [9]. Evaluation of IUVs is especially challenging because IUVs often taper to threadlike tributaries near the confluence with the internal iliac vein [9]. However, in exceptional cases, CTA may be superior to MRA for imaging SUVs/IUVs. For instance, we have observed a case of a vein running dorsally to the sigmoid colon, which was better visualized on CT due to an artifact that precluded evaluation on MRA.

In our experience, it is helpful to synchronize the data acquisition time with the cardiac cycle using electrocardiographic gating [11,12,13]. The tendency of MRA to produce artifacts caused by peristalsis and respiratory motion may be suppressed by applying several adjunct techniques [12], e.g., the amount of gas in the colon can be decreased by the administration of a mini-enema. Also, the motion artifacts of the colon can be decreased by anti-peristaltic agents, e.g., glucagon infusion. Respiratory motion artifacts may be decreased by prone-positioning of the donor. The Valsalva maneuver has not been shown to improve venous characterization [10].

### 4.2. Best Practices and Future Directions

The delivery and practice of high-quality healthcare are strongly dependent on effective communication [14,15]. At the beginning of the DUETS, the initial three cases of thrombosis led us to revise our protocol. The revision included an introduction of a thorough radiologic evaluation of the grafts’ vasculature. The initial absence of this attitude might have been caused by insufficient communication on the level of transplant surgeon-radiologist. Thus, we are convinced that the radiologist must focus on the detailed description of the parameters pertinent to the procedure’s success. The radiologist should comment on the details that could render the uterus marginal or unacceptable, such as the diameter and number of inflow and outflow vessels, including their tapering, normal or abnormal course, degree of atherosclerosis, etc. Throughout the process of drafting this paper, the significance of effective surgeon-radiologist communication was recognized as a significant take-home message.

We also feel that the consistency of radiologic performance is crucial. In our trial, a few studies were evaluated by a radiologist not dedicated to this area of expertise. The inconsistency of evaluation might have contributed to the acceptance of a graft that might have been otherwise rejected. For instance, in one case, the former CT description failed to describe the arteries as diminutive with very narrow ostia and underscored the degree of atherosclerosis. The graft was lost due to hypo-perfusion on postoperative day one. The histological examination of the explanted graft showed mild to moderate arteriosclerotic changes with some lumen narrowing. Considering the highly specialized nature of the assessment of uterine vasculature, we recommend consistent interpretation by a radiologist dedicated to interaction with the transplant team.

Our recommendation for exclusion criteria would be (1) structural factors (adhesions, fibroids, adenomyosis, etc.); (2) uterine arterial factors such as very small caliber, moderate to severe atherosclerosis, severe tortuosity, and a congenital variant of ovarian arterial supply (which may preclude harvest); (3) venous factors such as diminutive veins and multiple threadlike tributaries with an absence of quality vein.

### 4.3. Limitations of the Paper

There are several limitations of this study. Firstly, the evaluation of a cohort of uterine LD candidates was retrospective. Secondly, the methodology of both centers, as well as of DUETS itself, is not entirely consistent since the DUETS protocol changed throughout the trial. Three cases of graft thrombosis induced the change at the beginning of the trial. Also, the Czech protocol does not make use of MRA. Thirdly, the sample size was small, which limited our ability to show statistical significance in some areas. The absent or incomplete back-table reports hampered the radiological-surgical correlation. We admit that our grading system of uterine arteries and veins is arbitrary. We perceive it as a basis for a possibly improved grading system in the future

## 5. Conclusions

Radiologic imaging using CTA and MRA offers utility in evaluating the uterine blood supply of LD candidates. MRA may afford an initial evaluation for uterine donor candidates, but CTA should be undertaken for problem solving. While CTA is very valuable for assessing uterine arteries, its potential for evaluating uterine venous drainage is limited. If MRA is performed consistently according to a well-established protocol and assessed by a dedicated radiologist, it can provide reliable and precise information on uterine venous patterns, such as vessel size, taper, plethora, the dominant route of drainage, and fenestrations/duplications/multiplicity of the veins. It is highly recommended that the radiologist comment on the reliability and confidence of the imaging study. It cannot be over-emphasized that the most crucial aspect of surgical imaging is the necessity of high-quality communication between a surgeon and a radiologist. Accurate data collection from imaging exams and back-table reporting is advised for the continued study of outcomes with respect to vascular findings.

## Figures and Tables

**Figure 1 jcm-11-04626-f001:**
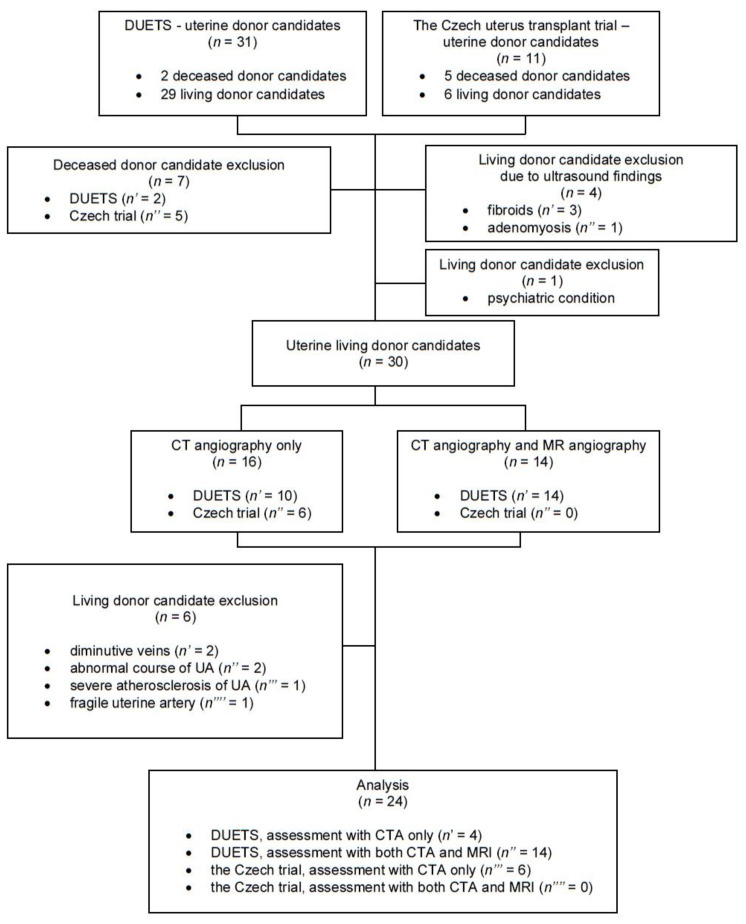
A flow chart depicting the inclusion and exclusion of living donor candidates into the study.

**Table 1 jcm-11-04626-t001:** The DUETS radiologic classification of uterine arteries, inferior and superior uterine veins.

Arteries					Commentary
Caliber	A (≥2 mm)	B (>1–2 mm)		C (≤1 mm)	
Tapering	Tapered			Not tapered	The artery is denominated tapered if it is thinner at its take-off from the IIA and gets more robust over its course to the uterus. The minus sign (−) is added to the grade of the caliber if the taper is present.
Atherosclerosis	None (0)	Mild (I)	Moderate (II)	Severe (III)	
Tortuosity	Mild (I)	Moderate (II)		Severe (III)	
Segment	Distance (mm)				The length of the segment between the posterior division of the IIA and the take-off of the UA is measured in mm.
Take-off and course	Normal			Abnormal	The take-off of the UA is considered normal if it takes off from the anterior division of the IIA.
**Veins**					**Commentary**
Caliber	A (≥6 mm)	B (≥4 and <6 mm)	C (<4 mm)	X (Vein not visualized)	
Tapering	Tapered	Non-tapered			If a vein is thinner at its confluence with the internal iliac vein (IIV) than over its course, we call it tapered. If a vein is tapered, the minus sign (−) is added to the grade of the caliber of the vein, e.g., B-.
Plethora	Plethora	No plethora			If para-uterine venous congestion is observed, the term plethora is used. In the case of plethora, no further description of the quality of the veins (caliber, fenestrations, tapering) is often possible.
Fenestrations, duplication, multiplicity	Fenestrated	Non-fenestrated			If IUV and/or SUV are not single, we distinguish fenestrated, duplicated, and multiple veins. A fenestrated vein is a single vein with two lumens (each lumen is divided from the other one by a septum). Duplicated and multiple veins are two or more separate veins running parallel to each other, respectively.
	Duplicated	Single			
	Multiple	Single			
Dominance	None or any of R IUV, L IUV, R SUV, L SUV				A dominant route of venous drainage is established. The dominance can be left-sided (left SUV and IUV), right-sided (right SUV and IUV), via both IUV (right and left IUV), etc.
Confidence	Very confident	Fairly confident		Not confident	The radiologist’s confidence in the quality of the imaging is graded. This is a very important characteristic of the evaluation. It helps surgeons understand how reliable the imaging and its description are. The confidence can be different for each vein separately.

Abbreviations: IIA, internal iliac artery; IUV, inferior uterine vein; L, left; R, right; SUV, superior uterine vein; UA, uterine artery.

**Table 2 jcm-11-04626-t002:** Correlation of a caliber of uterine artery on time from the last delivery.

The Caliber of Uterine Arteries	A (≥2 mm)	B (1–2 mm)	C (≤1 mm)
The number of arteries of a specified caliber (*n*)	10	26	12
The average time between the last delivery and donation (yr) ± SD	3.1 (±2.1 SD)	14.9 (±8.6 SD)	13.4 (±11.8 SD)
The range of time between the last delivery and donation (yr)	1.2–7	3–35	0.7–35
The average number of deliveries before donation ± SD	3.7 (±1.9 SD)	2.1 (±0.8 SD)	2.4 (±1.1 SD)
The range of number of deliveries before donation	2–7	0–3	1–4

Abbreviations: SD, standard deviation; yr, year(s).

**Table 3 jcm-11-04626-t003:** A comparison of CTA vs. MRA concerning the evaluation of quality and quantity of veins.

	Back-Table Finding	CTA Able to Confirm the Back-Table Finding	MRA Able to Confirm the Back-Table Finding
Fenestrated veins	7.1% (5/70)	0 (0/5)	20% (1/5)
Duplicated/multiple vein/plethora	20% (14/70)	21% (3/14)	41% (5/12)
A caliber of a vein	-	34% (24/70)	60% (26/44)
A number of veins	-	48% (34/70)	72% (32/44)

**Table 4 jcm-11-04626-t004:** The outcome and the caliber of uterine vessels.

The Outcome of the Graft	Arterial Thrombosis/Ischemia	Venous Thrombosis	Viable Graft/Ongoing Pregnancy	Live Birth
Number of grafts	4	2	5	13
Number (*n*) of A/A-minus arteries	0	0	33.3% (*n* = 4)	27.3% (*n* = 6)
Number (*n*) of B/B-minus arteries	62.5% (*n* = 5)	50% (*n* = 2)	50% (*n* = 6)	50% (*n* = 11)
Number (*n*) of C/C-minus arteries	37.5% (*n* = 3)	50% (*n* = 2)	16.7% (*n* = 2)	22.7% (*n* = 5)
Number (*n)* of A/A-minus veins	12.5% (*n* = 1)	20% (*n* = 1)	28.6% (*n* = 4)	22.2% (*n* = 6)
Number (*n)* of B/B-minus veins	50% (*n* = 4)	0	28.6% (*n* = 4)	55.6% (*n* = 15)
Number (*n)* of C/C-minus veins	37.5% (*n* = 3)	80% (*n* = 4)	21.4% (*n* = 3)	14.8% (*n* = 4)

## Data Availability

The data presented in this study are available on request from the corresponding author or L.J. The data are not publicly available due to confidentiality.

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
