# Peer review of "Radiologic Evaluation of Uterine Vasculature of Uterus Transplant Living Donor Candidates: DUETS Classification"

_jcm, 2022, doi:10.3390/jcm11154626_

Round 1

Reviewer 1 Report

Thanks for the nice paper. It gives an comprehensive and detailed overview of their experiences and a review of the literature concerning radiologic evaluation of uterine vasculature of uterus transplant living donor candidates. Of course, precise preoperative assessment of uterine vessels play a significant role in the choose of uterus transplant living donor candidates.

Suggestion”

1.        Should any previous medical history that affect thrombosis, such as smoking and history of systemic lupus erythematosus, should be emphasized.

2.        Which factor has a greater impact on the quality of transplanted uterine blood vessels, compared to the time between the last delivery and the time between menopause? Please discuss.

Author Response

Point 1: Any previous medical history that affect thrombosis, such as smoking and history of systemic lupus erythematosus, should be emphasized.

Response 1: We thank the reviewer for her/his kind words and helpful feedback. None of the included LDs suffered from systemic lupus erythematosus or any other systemic disease. Twenty-three of 24 (95,8%) LDs were non-smokers. One LD had a history of smoking (35 pack years). None of the recipients had any prothrombotic mutation or any medical history that affects thrombosis (lines 96-101).

Point 2: Which factor has a greater impact on the quality of transplanted uterine blood vessels, compared to the time between the last delivery and the time between menopause? Please discuss.

Response 2: We thank the reviewer for this very important question. This might be truly the key question for defining the optimal uterine donor candidates. However, we believe that our data set is too small to draw any conclusions on how significant the time factor is compared to other factors for the quality of uterine blood vessels. We are sorry we cannot provide more satisfying response but the amount of data we have is not sufficient to make definite conclusion on this. Indeed, many hysterectomized grafts had intraluminal obstruction in the form of fibrointimal hyperplasia or atherosclerosis (as was reported by the Swedish group, Dallas group or will be reported by the Czech team) but it is difficult to say what was the dominant causal factor, it might well be multifactorial (lines 249-250).

We appreciate the reviewer’s attention and helpful feedback.

Reviewer 2 Report

Kristek et al. report a retrospective dual center – Baylor college and Czech UTx – assessment of pre UTx vascular evaluation and ultimate UTx outcome.  The emphasize that the outflow of the uterine graft is provided by two to four venous anastomoses 47 using either superior uterine veins (SUV) and/or inferior uterine veins (IUV).  According to the authors, as much as 19.5% of uterus-transplanted patients underwent urgent graft hyster-24 ectomy due to thrombosis/hypoperfusion.

The authors state that 30 included candidates were evaluated with computed tomography angiography (CTA) with/without magnetic resonance angiography (MRA). Six out of 30 (20%) LD candidates 86 were excluded for various reasons (diminutive caliber of veins, severe atherosclerosis, abnormal course, or take-off of UA).  

Uterine artery and vein diameters were each rated in 3 categories, A to C based on CTA and MRA results. 

The authors’ results indicate that the total graft survival at 1 year was shown in 17 of 23 (73.9%) transplanted uteri. Six grafts of 23 (26.1%) were lost due to thrombosis. In 4 cases (8 arteries) that developed 190 arterial thrombosis/ischemia, the number of A/A-minus, B/B-minus, and C/C-minus arteries were 0 (n = 0), 62.5% (n = 5), and 37.5% (n = 3), respectively.

The authors conclude that uterine arterial factors such as very small caliber, moderate to severe atherosclerosis, severe tortuosity, a congenital variant of ovarian arterial supply (which may preclude harvest) and venous factors such as diminutive veins, multiple threadlike tributaries with an absence of quality vein should be considered as contraindication to UTx due to the increased risk of graft thrombosis.

The article proposed by Kristek et al. is of clear interest to specialists in UTx procedures.  The article is well written and soundly presented.

In my eyes however the authors ought to further compare their results with those of other groups.

Also, graft removal is reported as a very long procedures lasting up to 12h, while others report shorter surgical time.  The authors need to comment on the issue of surgical time – it differs according to reports between the two centers included in this study – and UTx outcome. 

Author Response

Point 1: The article proposed by Kristek et al. is of clear interest to specialists in UTx procedures.  The article is well written and soundly presented.

Response 1: We thank the reviewer for their kind words and helpful feedback.

Point 2: In my eyes however the authors ought to further compare their results with those of other groups.

Response 2: We thank the reviewer for their attention to this important point. Currently, only two studies are focusing on the preoperative radiologic evaluation of uterine vasculature. The study of Leonhardt et al. compares CTA vs. digital subtraction angiography vs. MRA for the visualization of uterine arteries of uterine LD candidates. They concluded that MRA should be the initial and the only modality in most cases. In a minority of cases when MRA fails to visualize the uterine arteries satisfactorily, CTA would be performed. Their study does not address the visualization of uterine venous drainage. We believe, as has already been published in the article of Mahmood et al., that CTA and MRA play rather complementary roles in the visualization of uterine arteries and veins. CTA provides a higher spatial resolution of uterine arteries and can detect calcified plaques more reliably. MRA, as we have shown, can yield more accurate results in the evaluation of uterine veins. We have added this commentary to the text of the manuscript (lines 257-267).

Point 3: Also, graft removal is reported as a very long procedures lasting up to 12h, while others report shorter surgical time.  The authors need to comment on the issue of surgical time – it differs according to reports between the two centers included in this study – and UTx outcome. 

Response 3: We thank the reviewer for bringing this to our attention. Uterine retrieval can be performed either open, laparoscopic, or robotic. All Czech LD uterine retrievals (6 cases) were performed open (median 5 h 51 min). In DUETS, open LD uterine retrievals were performed in 13 cases (median 6 h and 27 min), and the robotic uterine retrievals were performed in 8 cases (median 10 h and 46 min). Although robotic retrievals lasted longer than open procedures, no differences were observed in recipient outcomes (lines 251-256).
